# Trade-offs and Guarantees of Adversarial Representation Learning for Information Obfuscation

**Han Zhao**[*]
D. E. Shaw & Co.
`han.zhao@cs.cmu.edu`

**Jianfeng Chi**[*]
Department of Computer Science
University of Virginia
`jc6ub@virginia.edu`

**Yuan Tian**
Department of Computer Science
University of Virginia
`yuant@virginia.edu`

**Geoffrey J. Gordon**
Carnegie Mellon University
Microsoft Research Montreal
`geoff.gordon@microsoft.com`

## Abstract

Crowdsourced data used in machine learning services might carry sensitive information about attributes that users do not want to share. Various methods have been proposed to minimize the potential information leakage of sensitive attributes while maximizing the task accuracy. However, little is known about the theory behind these methods. In light of this gap, we develop a novel theoretical framework for attribute obfuscation. Under our framework, we propose a minimax optimization formulation to protect the given attribute and analyze its inference guarantees against worst-case adversaries. Meanwhile, it is clear that in general there is a tension between minimizing information leakage and maximizing task accuracy. To understand this, we prove an information-theoretic lower bound to precisely characterize the fundamental trade-off between accuracy and information leakage. We conduct experiments on two real-world datasets to corroborate the inference guarantees and validate this trade-off. Our results indicate that, among several alternatives, the adversarial learning approach achieves the best trade-off in terms of attribute obfuscation and accuracy maximization.

## 1 Introduction

With the growing demand for machine learning systems provided as services, a massive amount of data containing sensitive information, such as race, income level, age, etc., are generated and collected from local users. This poses a substantial challenge and it has become an imperative object of study in machine learning [18], computer vision [6, 34], healthcare [2, 3], speech recognition [30], and many other domains. In this paper, we consider a practical scenario where the prediction vendor requests crowdsourced data for a target task, e.g, scientific modeling. The data owner agrees on the data usage for the target task while she does not want her other sensitive information (*e.g.*, age, race) to be leaked. The goal in this context is then to obfuscate sensitive attributes of the sanitized data released by data owner from potential attribute inference attacks from a malicious adversary. For example, in an online advertising scenario, while the user (data owner) may agree to share her historical purchasing events, she also wants to protect her age information so that no malicious adversary can infer her age range from the shared data. Note that simply removing age attribute

---

[*]The first two authors contributed equally to this work. Work done while HZ was at Carnegie Mellon University.

from the shared data is insufficient for this purpose, due to the redundant encoding in data, i.e., other attributes may have a high correlation with age.

Under this scenario, a line of work [4, 14, 19, 22, 24, 25, 32–34] aims to address the problem in the framework of (constrained) minimax problem. However, the theory behind these methods is little known. Such a gap between theory and practice calls for an important and appealing challenge:

> *Can we prevent the information leakage of the sensitive attribute while still maximizing the task accuracy? Furthermore, what is the fundamental trade-off between attribute obfuscation and accuracy maximization in the minimax problem?*

Under the setting of attribute obfuscation, the notion of information confidentiality should be attribute-specific: the goal is to protect specific attributes from being inferred by malicious adversaries as much as possible. Note that this is in sharp contrast with differential privacy (we systematically compare the related notions in Sec. 5 Related Work), where mechanisms are usually designed to resist worst-case membership query among all the data owners instead of preventing information leakage of the sensitive attribute [11]. From this perspective, our relaxed definition of attribute obfuscation against adversaries also allows for a more flexible design of algorithms with better accuracy.

**Our Contributions**   In this paper, we first formally define the notion of attribute inference attack in our setting and justify why our definitions are particularly suited under our setting. Through the lens of representation learning, we formulate the problem of accuracy maximization with information obfuscation constraint as a minimax optimization problem. To provide a formal guarantee on attribute obfuscation, we prove an information-theoretic lower bound on the inference error of the protected attribute under attacks from arbitrary adversaries. To investigate the relationship between attribute obfuscation and accuracy maximization, we also prove a theorem that formally characterizes the inherent trade-off between these two concepts. We conduct experiments to corroborate our formal guarantees and validate the inherent trade-offs in different attribute obfuscation algorithms. From our empirical results, we conclude that the adversarial representation learning approach achieves the best trade-off in terms of attribute obfuscation and accuracy maximization, among various state-of-the-art attribute obfuscation algorithms.

## 2   Preliminaries

**Problem Setup**   We focus on the setting where the goal of the adversary is to perform *attribute inference*. This setting is ubiquitous in sever-client paradigm where machine learning is provided as a service (MLaaS, Ribeiro et al. [27]). Formally, there are two parties in the system, namely the prediction vendor and the data owner. We consider the practical scenarios where users agree to contribute their data for specific purposes (*e.g.*, training a machine learning model) but do not want others to infer their sensitive attributes in the data, such as health information, race, gender, etc. The prediction vendor will not collect raw user data but processed user data and the target attribute for the target task. In our setting, we assume the adversary cannot get other auxiliary information than the processed user data. In this case, the adversary can be anyone who can get access to the processed user data to some extent and wants to infer other private information. For example, malicious machine learning service providers are motivated to infer more information from users to do user profiling and targeted advertisements. The goal of the data owner is to provide as much information as possible to the prediction vendor to maximize the vendor's own accuracy, but under the constraint that the data owner should also protect the private information of the data source, *i.e.*, *attribute obfuscation*. For ease of discussion, in our following analysis, we assume the the prediction vendor performs binary classification on the processed data. Extensions to multi-class classification is straightforward.

**Notation**   We use $\mathcal{X}$, $\mathcal{Y}$ and $\mathcal{A}$ to denote the input, output and adversary's output space, respectively. Accordingly, we use $X, Y, A$ to denote the random variables which take values in $\mathcal{X}, \mathcal{Y}$ and $\mathcal{A}$. We note that in our framework the input space $\mathcal{X}$ may or may not contain the sensitive attribute $A$. For two random variables $X$ and $Y$, $I(X;Y)$ denotes the mutual information between $X$ and $Y$. We use $H(X)$ to mean the Shannon entropy of random variable $X$. Similarly, we use $H(X \mid Y)$ to denote the conditional entropy of $X$ given $Y$. We assume there is a joint distribution $\mathcal{D}$ over $\mathcal{X} \times \mathcal{Y} \times \mathcal{A}$ from which the data are sampled. To make our notation consistent, we use $\mathcal{D}_{\mathcal{X}}$, $\mathcal{D}_{\mathcal{Y}}$ and $\mathcal{D}_{\mathcal{A}}$ to denote the marginal distribution of $\mathcal{D}$ over $\mathcal{X}$, $\mathcal{Y}$ and $\mathcal{A}$. Given a feature map function $f : \mathcal{X} \rightarrow \mathcal{Z}$ that maps instances from the input space $\mathcal{X}$ to feature space $\mathcal{Z}$, we define

$\mathcal{D}^f := \mathcal{D} \circ f^{-1}$ to be the induced (pushforward) distribution of $\mathcal{D}$ under $f$, i.e., for any event $E' \subseteq \mathcal{Z}$, $\Pr_{\mathcal{D}^f}(E') := \Pr_{\mathcal{D}}(\{x \in \mathcal{X} \mid f(x) \in E'\})$.

To simplify the exposition, we mainly discuss the setting where $\mathcal{X} \subseteq \mathbb{R}^d, \mathcal{Y} = \mathcal{A} = \{0, 1\}$, but the underlying theory and methodology could easily be extended to the categorical case as well. In what follows, we first formally define both the *accuracy* of the prediction vendor for the individualized service and the *attribute inference advantage* of an adversary. It is worth pointing out that our definition of inference advantage is *attribute-specific*. In particular, we seek to keep the data useful while being robust to an adversary on protecting specific attribute information from attack.

A *hypothesis* is a function $h : \mathcal{X} \to \mathcal{Y}$. The *error* of a hypothesis $h$ under the distribution $\mathcal{D}$ over $\mathcal{X} \times \mathcal{Y}$ is defined as: $\mathrm{Err}(h) := \mathbb{E}_{\mathcal{D}}[|Y - h(X)|]$. Similarly, we use $\widehat{\mathrm{Err}}(h)$ to denote the empirical error of $h$ on a sample from $\mathcal{D}$. For binary classification problem, when $h(\mathbf{x}) \in \{0, 1\}$, the above loss also reduces to the error rate of classification. Let $\mathcal{H}$ be the space of hypotheses. In the context of binary classification, we define the accuracy of a hypothesis $h \in \mathcal{H}$ as:

**Definition 2.1** (Accuracy). The accuracy of $h \in \mathcal{H}$ is $\mathrm{ACC}(h) := 1 - \mathbb{E}_{\mathcal{D}}[|Y - h(X)|]$.

For binary classification, we always have $0 \leq \mathrm{ACC}(h) \leq 1$, $\forall h \in \mathcal{H}$. Similarly, an *adversarial hypothesis* is a function of $h_A : \mathcal{X} \to \mathcal{A}$. Next we define a measure of how much advantage of attribute inference gained from a particular attack space in our framework:

**Definition 2.2** (Attribute Inference Advantage). The inference advantage w.r.t. attribute $A$ under attacks from $\mathcal{H}_A$ is defined as $\mathrm{ADV}(\mathcal{H}_A) := \max_{h_A \in \mathcal{H}_A} |\Pr_{\mathcal{D}}(h_A(X) = 1 \mid A = 1) - \Pr_{\mathcal{D}}(h_A(X) = 1 \mid A = 0)|$.

Again, it is straightforward to verify that $0 \leq \mathrm{ADV}(\mathcal{H}_A) \leq 1$. Based on our definition, $\mathrm{ADV}(\mathcal{H}_A)$ then measures maximal inference advantage that the adversary in $\mathcal{H}_A$ can gain. We can also refine the above definition to a particular hypothesis $h_A : \mathcal{X} \to \{0, 1\}$ to measure its ability to steal information about $A$: $\mathrm{ADV}(h_A) = |\Pr_{\mathcal{D}}(h_A(X) = 1 \mid A = 1) - \Pr_{\mathcal{D}}(h_A(X) = 1 \mid A = 0)|$.

**Proposition 2.1.** Let $h_A : \mathcal{X} \to \{0, 1\}$ be a hypothesis, then $\mathrm{ADV}(h_A) = 0$ iff $I(h_A(X); A) = 0$ and $\mathrm{ADV}(h_A) = 1$ iff $h_A(X) = A$ almost surely or $h_A(X) = 1 - A$ almost surely.

Proposition 2.1 justifies Definition 2.2 on how well an adversary $h_A$ can infer about $A$ from $X$: when $\mathrm{ADV}(h_A) = 0$, it means that $h_A(X)$ contains no information about the sensitive attribute $A$. On the other hand, if $\mathrm{ADV}(h_A) = 1$, then $h_A(X)$ fully predicts $A$ (or equivalently, $1 - A$) from input $X$. In the latter case $h_A(X)$ also contains perfect information of $A$ in the sense that $I(h_A(X); A) = H(A)$, i.e., the Shannon entropy of $A$. It is worth pointing out that Definition 2.2 is insensitive to the marginal distribution of $A$, and hence is more robust than other definitions such as the error rate of predicting $A$. In that case, if $A$ is extremely imbalanced, even a naive predictor can attain small prediction error by simply outputting constant. We call a hypothesis space $\mathcal{H}_A$ *symmetric* if $\forall h_A \in \mathcal{H}_A, 1 - h_A \in \mathcal{H}_A$ as well. When $\mathcal{H}_A$ is symmetric, we can also relate $\mathrm{ADV}(\mathcal{H}_A)$ to a binary classification problem:

**Proposition 2.2.** If $\mathcal{H}_A$ is symmetric, then $\mathrm{ADV}(\mathcal{H}_A) + \min_{h_A \in \mathcal{H}_A} \Pr(h_A(X) = 0 \mid A = 1) + \Pr(h_A(X) = 1 \mid A = 0) = 1$.

Consider the confusion matrix between the actual sensitive attribute $A$ and its predicted variable $h_A(X)$. The false positive rate (eqv. Type-I error) is defined as FPR = FP / (FP + TN) and the false negative rate (eqv. Type-II error) is similarly defined as FNR = FN / (FN + TP). Using the terminology of confusion matrix, it is then clear that $\Pr(h_A(X) = 0 \mid A = 1) = $ FNR and $\Pr(h_A(X) = 1 \mid A = 0) = $ FPR. In other words, Proposition 2.2 says that if $\mathcal{H}_A$ is symmetric, then the larger the attribute inference advantage of $\mathcal{H}_A$, the smaller the minimum sum of Type-I and Type-II error under attacks from $\mathcal{H}_A$.

## 3 Main Results

Given a set of samples $\mathbf{S} = \{(\mathbf{x}_i, y_i, a_i)\}_{i=1}^n$ drawn i.i.d. from the joint distribution $\mathcal{D}$, how can the data owner keep the data useful while keeping the sensitive attribute $A$ obfuscated under potential attacks from malicious adversaries? Through the lens of representation learning, we seek to find a (non-linear) feature representation $f : \mathcal{X} \to \mathcal{Z}$ from input space $\mathcal{X}$ to feature space $\mathcal{Z}$ such that $f$

still preserves relevant information w.r.t. the target task of inferring $Y$ while hiding sensitive attribute $A$. Specifically, we can solve the following unconstrained regularized problem with $\lambda > 0$:

$$\min_{h \in \mathcal{H}, f} \max_{h_A \in \mathcal{H}_A} \widehat{\mathrm{Err}}(h \circ f) - \lambda\big(\Pr_{\mathsf{S}}(h_A(f(X)) = 0 \mid A = 1) + \Pr_{\mathsf{S}}(h_A(f(X)) = 1 \mid A = 0)\big) \tag{1}$$

It is worth pointing out that the optimization formulation in (1) admits an interesting game-theoretic interpretation, where two agents $f$ and $h_A$ play a game whose score is defined by the objective function in (1). Intuitively, $h_A$ seeks to minimize the sum of Type-I and Type-II error while $f$ plays against $h_A$ by learning transformation to removing information about the sensitive attribute $A$. Algorithmically, for the data owner to achieve the goal of hiding information about the sensitive attribute $A$ from malicious adversary, it suffices to learn a representation that is independent of $A$:

**Proposition 3.1.** Let $f : \mathcal{X} \to \mathcal{Z}$ be a deterministic function and $\mathcal{H}_A \subseteq 2^{\mathcal{Z}}$ be a hypothesis class over $\mathcal{Z}$. For any joint distribution $\mathcal{D}$ over $X, A, Y$, if $I(f(X); A) = 0$, then $\mathrm{ADV}(\mathcal{H}_A \circ f) = 0$.

Note that in this sequential game, $f$ is the first-mover and $h_A$ is the second. Hence without explicit constraint $f$ possesses a first-mover advantage so that $f$ can dominate the game by simply mapping all the input $X$ to a constant or uniformly random noise[2]. To avoid these degenerate cases, the first term in the objective function of (1) acts as an incentive to encourage $f$ to preserve task-related information. But will this incentive compromise the information of $A$? As an extreme case if the target variable $Y$ and the sensitive attribute $A$ are perfectly correlated, then it should be clear that there is a trade-off in achieving accuracy and preventing information leakage of the attribute. In Sec. 3.2 we shall provide an information-theoretic bound to precisely characterize such trade-off.

## 3.1 Formal Guarantees against Attribute Inference

In the unconstrained minimax formulation (1), the hyperparameter $\lambda$ measures the trade-off between accuracy and information obfuscation. On one hand, if $\lambda \to 0$, we barely care about the information obfuscation of $A$ and devote all the focus to maximize our accuracy. On the other extreme, if $\lambda \to \infty$, we are only interested in obfuscating the sensitive information. In what follows we analyze the true error that an optimal adversary has to incur in the limit when both the task classifier and the adversary have unlimited capacity, i.e., they can be any randomized functions from $\mathcal{Z}$ to $\{0, 1\}$. To study the true error, we hence use the population loss rather than the empirical loss in our objective function. Furthermore, since the binary classification error in (1) is NP-hard to optimize even for hypothesis class of linear predictors, in practice we consider the cross-entropy loss function as a convex surrogate loss. With a slight abuse of notation, the cross-entropy loss $\mathrm{CE}_Y(h)$ of a probabilistic hypothesis $h : \mathcal{X} \to [0, 1]$ w.r.t. $Y$ on a distribution $\mathcal{D}$ is defined as follows:

$$\mathrm{CE}_Y(h) := -\mathbb{E}_{\mathcal{D}}[\mathbb{I}(Y = 0) \log(1 - h(X)) + \mathbb{I}(Y = 1) \log(h(X))].$$

We also use $\mathrm{CE}_A(h_A)$ to mean the cross-entropy loss of the adversary $h_A$ w.r.t. $A$. Using the same notation, the optimization formulation with cross-entropy loss becomes:

$$\min_{h \in \mathcal{H}, f} \max_{h_A \in \mathcal{H}_A} \mathrm{CE}_Y(h \circ f) - \lambda \cdot \mathrm{CE}_A(h_A \circ f) \tag{2}$$

Given a feature map $f : \mathcal{X} \to \mathcal{Z}$, assume that $\mathcal{H}$ contains all the possible probabilistic classifiers from the feature space $\mathcal{Z}$ to $[0, 1]$. For example, a probabilistic classifier can be constructed by first defining a function $h : \mathcal{Z} \to [0, 1]$ followed by a random coin flipping to determine the output label, where the probability of the coin being 1 is given by $h(Z)$. Under such assumptions, the following lemma shows that the optimal target classifier under $f$ is given by the conditional distribution $h^*(Z) := \Pr(Y = 1 \mid Z)$.

**Lemma 3.1.** For any feature map $f : \mathcal{X} \to \mathcal{Z}$, assume that $\mathcal{H}$ contains all the probabilistic classifiers, then $\min_{h \in \mathcal{H}} \mathrm{CE}_Y(h \circ f) = H(Y \mid Z)$ and $h^*(Z) := \arg\min_{h \in \mathcal{H}} \mathrm{CE}_Y(h \circ f) = \Pr(Y = 1 \mid Z = f(X))$.

By a symmetric argument, we can also see that the worst-case (optimal) adversary under $f$ is the conditional distribution $h_A^*(Z) := \Pr(A = 1 \mid Z)$ and $\min_{h_A \in \mathcal{H}_A} \mathrm{CE}_A(h_A \circ f) = H(A \mid Z)$.

Hence we can further simplify the optimization formulation (2) to the following form where the only optimization variable is the feature map $f$:

$$\min_f \quad H(Y \mid Z = f(X)) - \lambda H(A \mid Z = f(X)) \tag{3}$$

Since $Z = f(X)$ is a deterministic feature map, it follows from the basic properties of Shannon entropy that

$$H(Y \mid X) \leq H(Y \mid Z = f(X)) \leq H(Y), \qquad H(A \mid X) \leq H(A \mid Z = f(X)) \leq H(A)$$

which means that $H(Y \mid X) - \lambda H(A)$ is a lower bound of the optimum of the objective function in (3). However, such lower bound is not necessarily achievable. To see this, consider the simple case where $Y = A$ almost surely. In this case there exists no deterministic feature map $Z = f(X)$ that is both a sufficient statistics of $X$ w.r.t. $Y$ while simultaneously filters out all the information w.r.t. $A$ except in the degenerate case where $A(Y)$ is constant. Next, to show that solving the optimization problem in (3) helps to remove sensitive information, the following theorem gives a bound of attribute inference in terms of the error that has to be incurred by the optimal adversary:

**Theorem 3.1.** Let $f^*$ be the optimal feature map of (3) and define $H^* := H(A \mid Z = f^*(X))$. Then for any adversary $\widehat{A}$ such that $I(\widehat{A}; A \mid Z) = 0$, $\Pr_{\mathcal{D}^{f^*}}(\widehat{A} \neq A) \geq H^*/2\lg(6/H^*)$.

**Remark**   Theorem 3.1 shows that whenever the conditional entropy $H^* = H(A \mid Z = f^*(X))$ is large, then the inference error of the protected attribute incurred by any (randomized) adversary has to be at least $\Omega(H^*/\log(1/H^*))$. The assumption $I(\widehat{A}; A \mid Z) = 0$ says that, given the processed feature $Z$, the adversary $\widehat{A}$ could not use any external information that depends on the true sensitive attribute $A$. As we have already shown above, the conditional entropy essentially corresponds to the second term in our objective function, whose optimal value could further be flexibly adjusted by tuning the trade-off parameter $\lambda$. As a final note, Theorem 3.1 also shows that representation learning helps to remove the information about $A$ since we always have $H(A \mid Z = f(X)) \geq H(A \mid X)$ for any deterministic feature map $f$ so that the lower bound of inference error by any adversary is larger after learning the representation $Z = f(X)$.

### 3.2   Inherent trade-off between Accuracy and Attribute Obfuscation

In this section we shall provide an information-theoretic bound to quantitatively characterize the inherent trade-off between these accuracy maximization and attribute obfuscation, due to the discrepancy between the conditional distributions of the target variable given the sensitive attribute. Our result is algorithm-independent, hence it applies to a general setting where there is a need to preserve both terms. To the best of our knowledge, this is the first information-theoretic result to precisely quantify such trade-off. Due to space limit, we defer all the proofs to the appendix.

Before we proceed, we first define several information-theoretic concepts that will be used in our analysis. For two distributions $\mathcal{D}$ and $\mathcal{D}'$, the Jensen-Shannon (JS) divergence $D_{JS}(\mathcal{D}, \mathcal{D}')$ is: $D_{JS}(\mathcal{D}, \mathcal{D}') := \frac{1}{2}D_{KL}(\mathcal{D} \parallel \mathcal{D}_M) + \frac{1}{2}D_{KL}(\mathcal{D}' \parallel \mathcal{D}_M)$, where $D_{KL}(\cdot \parallel \cdot)$ is the Kullback–Leibler (KL) divergence and $\mathcal{D}_M := (\mathcal{D} + \mathcal{D}')/2$. The JS divergence can be viewed as a symmetrized and smoothed version of the KL divergence, However, unlike the KL divergence, the JS divergence is bounded: $0 \leq D_{JS}(\mathcal{D}, \mathcal{D}') \leq 1$. Additionally, from the JS divergence, we can define a distance metric between two distributions as well, known as the JS distance [13]: $d_{JS}(\mathcal{D}, \mathcal{D}') := \sqrt{D_{JS}(\mathcal{D}, \mathcal{D}')}$. With respect to the JS distance, for any feature space $\mathcal{Z}$ and any deterministic mapping $f : \mathcal{X} \to \mathcal{Z}$, we can prove the following lemma via the celebrated data processing inequality:

**Lemma 3.2.** Let $\mathcal{D}_0$ and $\mathcal{D}_1$ be two distributions over $\mathcal{X}$ and let $\mathcal{D}_0^f$ and $\mathcal{D}_1^f$ be the induced distributions of $\mathcal{D}_0$ and $\mathcal{D}_1$ over $\mathcal{Z}$ by function $f$, then $d_{JS}(\mathcal{D}_0^f, \mathcal{D}_1^f) \leq d_{JS}(\mathcal{D}_0, \mathcal{D}_1)$.

Without loss of generality, any method aiming to predict the target variable $Y$ defines a Markov chain as $X \xrightarrow{f} Z \xrightarrow{h} \hat{Y}$, where $\hat{Y}$ is the predicted target variable given by hypothesis $h$ and $Z$ is the intermediate representation defined by the feature mapping $f$. Hence for any distribution $\mathcal{D}_0(\mathcal{D}_1)$ of $X$, this Markov chain also induces a distribution $\mathcal{D}_0^{h \circ f}(\mathcal{D}_1^{h \circ f})$ of $\hat{Y}$ and a distribution $\mathcal{D}_0^f(\mathcal{D}_1^f)$ of $Z$. Now let $\mathcal{D}_0^Y(\mathcal{D}_1^Y)$ be the underlying true conditional distribution of $Y$ given $A = 0(A = 1)$. Realize

that the JS distance is a metric, the following chain of triangular inequalities holds:

$$d_{\text{JS}}(\mathcal{D}_0^Y, \mathcal{D}_1^Y) \leq d_{\text{JS}}(\mathcal{D}_0^Y, \mathcal{D}_0^{h \circ f}) + d_{\text{JS}}(\mathcal{D}_0^{h \circ f}, \mathcal{D}_1^{h \circ f}) + d_{\text{JS}}(\mathcal{D}_1^{h \circ f}, \mathcal{D}_1^Y).$$

Combining the above inequality with Lemma 3.2 to show $d_{\text{JS}}(\mathcal{D}_0^{h \circ f}, \mathcal{D}_1^{h \circ f}) \leq d_{\text{JS}}(\mathcal{D}_0^f, \mathcal{D}_1^f)$, we immediately have:

$$d_{\text{JS}}(\mathcal{D}_0^Y, \mathcal{D}_1^Y) \leq d_{\text{JS}}(\mathcal{D}_0^Y, \mathcal{D}_0^{h \circ f}) + d_{\text{JS}}(\mathcal{D}_0^f, \mathcal{D}_1^f) + d_{\text{JS}}(\mathcal{D}_1^{h \circ f}, \mathcal{D}_1^Y).$$

Intuitively, $d_{\text{JS}}(\mathcal{D}_0^Y, \mathcal{D}_0^{h \circ f})$ and $d_{\text{JS}}(\mathcal{D}_1^Y, \mathcal{D}_1^{h \circ f})$ measure the distance between the predicted and the true target distribution on $A = 0/1$ cases, respectively. Formally, let $\text{Err}_a(h \circ f)$ be the prediction error of function $h \circ f$ conditioned on $A = a$. With the help of Lemma A.2, the following result establishes a relationship between $d_{\text{JS}}(\mathcal{D}_a^Y, \mathcal{D}_a^{h \circ f})$ and the accuracy of $h \circ f$:

**Lemma 3.3.** Let $\hat{Y} = h(f(X)) \in \{0,1\}$ be the predictor, then for $a \in \{0,1\}$, $d_{\text{JS}}(\mathcal{D}_a^Y, \mathcal{D}_a^{h \circ f}) \leq \sqrt{\text{Err}_a(h \circ f)}$.

Combine Lemma 3.2 and Lemma 3.3, we get the following key lemma that is the backbone for proving the main results in this section:

**Lemma 3.4** (Key lemma). Let $\mathcal{D}_0, \mathcal{D}_1$ be two distributions over $\mathcal{X} \times \mathcal{Y}$ conditioned on $A = 0$ and $A = 1$ respectively. Assume the Markov chain $X \xrightarrow{f} Z \xrightarrow{h} \hat{Y}$ holds, then $\forall h \in \mathcal{H}$:

$$d_{\text{JS}}(\mathcal{D}_0^Y, \mathcal{D}_1^Y) \leq \sqrt{\text{Err}_0(h \circ f)} + d_{\text{JS}}(\mathcal{D}_0^f, \mathcal{D}_1^f) + \sqrt{\text{Err}_1(h \circ f)}.$$

We emphasize that for $a \in \{0,1\}$, the term $\text{Err}_a(h \circ f)$ measures the conditional error of the predicted variable $\hat{Y}$ by the composite function $h \circ f$ over $\mathcal{D}_a$. Similarly, we can define the *conditional accuracy* for $a \in \{0,1\}$ : $\text{ACC}_a(h \circ f) := 1 - \text{Err}_a(h \circ f)$. The following main theorem then characterizes a fundamental trade-off between accuracy and attribute obfuscation:

**Theorem 3.2.** Let $\mathcal{H}_A \subseteq 2^{\mathcal{Z}}$ be the hypothesis space of all the classifiers from $\mathcal{Z}$ to $\{0,1\}$. Assume the conditions in Lemma 3.4 hold, then $\forall h \in \mathcal{H}$, $\text{ACC}_0(h \circ f) + \text{ACC}_1(h \circ f) \leq 2 - \frac{1}{3} D_{\text{JS}}(\mathcal{D}_0^Y, \mathcal{D}_1^Y) + \text{ADV}(\mathcal{H}_A \circ f)$.

The upper bound given in Theorem 3.2 shows that when the marginal distribution of the target variable $Y$ differ between two cases $A = 0$ or $A = 1$, then it is impossible to perfectly maximize accuracy and prevent the sensitive attribute being inferred. Furthermore, the trade-off due to the difference in marginal distributions is precisely given by the JS divergence $D_{\text{JS}}(\mathcal{D}_0^Y, \mathcal{D}_1^Y)$. Next, if we would like to decrease the advantage of adversaries, $\text{ADV}(\mathcal{H}_A \circ f)$, through learning proper feature transformation $f$, then the upper bound on the sum of conditional accuracy also becomes smaller, for any predictor $h$. Note that in Theorem 3.2 the upper bound holds for *any* adversarial hypothesis $h_A$ in the richest hypothesis class $\mathcal{H}_A$ that contains all the possible binary classifiers. Put it another way, if we would like to maximally obfuscate information w.r.t. sensitive attribute $A$, then we have to incur a large joint error:

**Theorem 3.3.** Assume the conditions in Theorem 3.2 hold. If $\text{ADV}(\mathcal{H}_A \circ f) \leq D_{\text{JS}}(\mathcal{D}_0^Y, \mathcal{D}_1^Y)$, then $\forall h \in \mathcal{H}$, $\text{Err}_0(h \circ f) + \text{Err}_1(h \circ f) \geq \frac{1}{2}\left(d_{\text{JS}}(\mathcal{D}_0^Y, \mathcal{D}_1^Y) - \sqrt{\text{ADV}(\mathcal{H}_A \circ f)}\right)^2$.

**Remark**   The above lower bound characterizes a fundamental trade-off between information obfuscation of the sensitive attribute and joint error of target task. In particular, up to a certain level $D_{\text{JS}}(\mathcal{D}_0^Y, \mathcal{D}_1^Y)$, the larger the inference advantage that the adversary can gain, the smaller the joint error. In light of Proposition 3.1, this means that although the data-owner, or the first-mover $f$, could try to maximally filter out the sensitive information via constructing $f$ such that $f(X)$ is independent of $A$, such construction will also inevitably compromise the joint accuracy of the prediction vendor. It is also worth pointing out that our results in both Theorem 3.2 and Theorem 3.3 are attribute-independent in the sense that neither of the bounds depends on the marginal distribution of $A$. Instead, all the terms in our results only depend on the conditional distributions given $A = 0$ and $A = 1$. This is often more desirable than bounds involving mutual information, *e.g.*, $I(A, Y)$, since $I(A, Y)$ is close to 0 if the marginal distribution of $A$ is highly imbalanced.

# 4 Experiments

In this section, we conduct experiments to investigate the following questions:

**Q1** Are our formal guarantees valid for different attribute obfuscation methods and the inherent trade-offs between attribute information obfuscation and accuracy maximization exist in all methods?

**Q2** Which attribute obfuscation algorithms achieve the best trade-offs in terms of attribute obfuscation and accuracy maximization?

## 4.1 Datasets and Setup

In our experiments, we use: (1) Adult dataset [8]: The Adult dataset is a benchmark dataset for classification. The task is to predict whether an individual's income is greater or less than 50K/year based on census data. In our experiment we set the target task as income prediction and the malicious task done by the adversary as inferring gender, age and education, respectively. (2) UTKFace dataset [38]: The UTKFace dataset is a large-scale face benchmark dataset containing more than 20,000 images with annotations of age, gender, and ethnicity. In our experiment, we set our target task as gender classification and we use the age and ethnicity as the protected attributes. We refer readers to Sec. C in the appendix for detailed descriptions about the data pre-processing pipeline and the data distribution for each dataset.

We conduct experiments with the following methods to verify our theoretical results and provide a thorough practical comparison among these methods. 1). Privacy Partial Least Squares (PPLS) [14], 2). Privacy Linear Discriminant Analysis (PLDA) [33], 3). Minimax filter with alternative update (ALT-UP) [19], 4) Maximum Entropy Adversarial Representation Learning (MAX-ENT) [28] 5). Gradient Reversal Layer (GRL) [17] 6). Principal Component Analysis (PCA) 7). Normal Training (NORM-TRAIN), 8) Local Differential Privacy (LDP) with Laplacian mechanism, 9). Differentially Private SGD (DPSGD) [1]. Among the first seven methods, the first five are state-of-the-art minimax methods for protecting against attribute inference attacks while the latter two are normal representation learning baselines for comprehensive comparison. Although DP is not tailored to attribute obfuscation, we can still add two DP baselines to examine the accuracy and attribute obfuscation trade-off for comparison[3]. To ensure the comparison is fair among different methods, we conduct a controlled experiment by using the same network structure as the baseline hypothesis among all the methods for each dataset. For each experiment on the Adult dataset and UTKFace dataset, we repeat the experiments for ten times to report both the average performance and their standard deviations. Sec. C in the appendix provides detailed descriptions about the methods and the hyperparameter settings.

Note that in practice due to the non-convex nature of optimizing deep neural nets, we cannot guarantee to find the global optimal conditional entropy $H^*$. Hence in order to compute the formal guarantee given by our lower bound in Theorem 3.1, we use the cross-entropy loss of the optimal adversary found by our algorithm on inferring the sensitive attribute $A$. Furthermore, since our analysis only applies to representation learning based approaches, we do not have similar guarantee for DP-related methods in our context. We visualize the performances of the aforementioned algorithms on attribute obfuscation and accuracy maximization in Figure 1 and Figure 2, respectively.

## 4.2 Results and Analysis

**Validation of Our Theory (Q1)**    From Figure 1, we can see that the formal guarantees are valid for all representation learning approaches. With the results in Figure 2, we also see that no methods are perfect in both achieving both attribute obfuscation and accuracy maximization: the methods with small accuracy loss comes with relative low inference errors and vice versa.

**Comparison with Different Methods (Q2)**    Among all methods, LDP, PLDA, ALT-UP, MAX-ENT and GRL are effective in attribute obfuscation by forcing the optimal adversary to incur a large inference error in Figure 1. On the other hand, PCA and NORM-TRAIN are the least effective ones. This is expected as neither NORM-TRAIN nor PCA filters information in data about the sensitive attribute $A$.

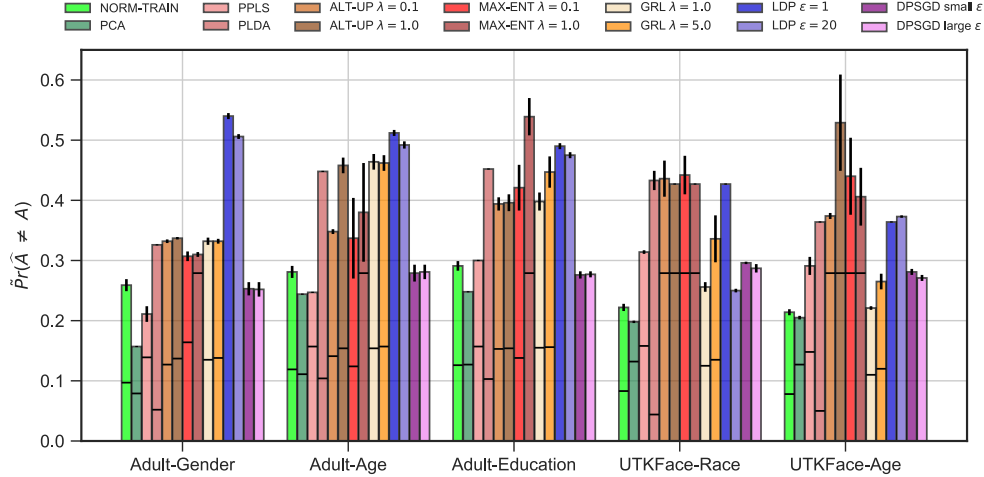

Figure 1: Performance on attribute obfuscation of different methods (the larger the better). The horizontal lines across the bars indicate the corresponding formal guarantees given by our lower bound in Theorem 3.1.

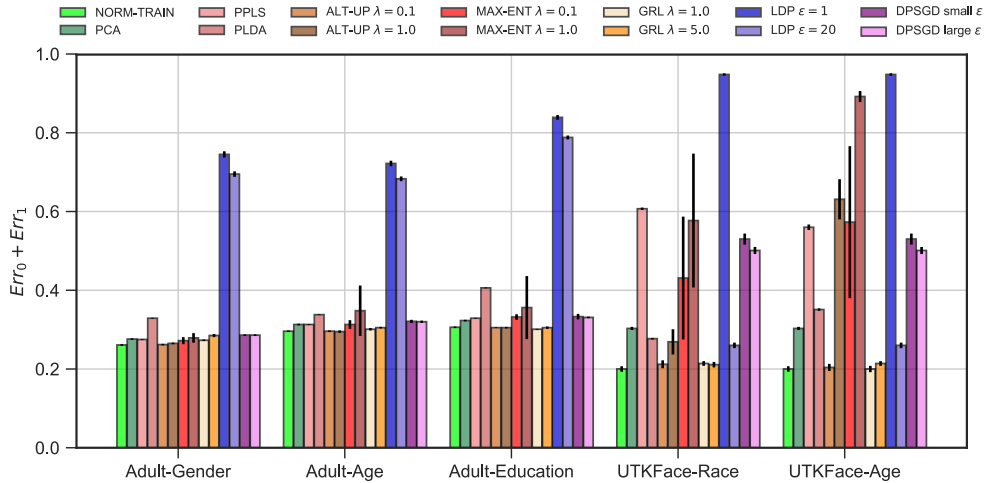

Figure 2: The joint conditional error ($\text{Err}_0 + \text{Err}_1$, the smaller the better) of different methods.

From Figure 2, we can also see a sharp contrast between DP-based methods and other methods in terms of the joint conditional error on the target task: both LDP and DPSGD could incur significant accuracy loss compared with other methods. Combining this one with our previous observation from Figure 1, we can see that DP-based methods either make data private by adding large amount of noise to filter out both target-related information and sensitive-related information available in the data, or add insufficient amount of noise so that both target-related and sensitive-related information is well preserved. As a comparison, representation learning based approaches leads to a better trade-off.

Among the representation learning methods, PLDA, ALT-UP, MAX-ENT and GRL perform the best in attribute obfuscation. Compared to PLDA and GRL, ALT-UP and MAX-ENT incur significant drops in accuracy when $\lambda$ is large. It is also worth to note that different adversarial representation learning methods have different sensitivity on $\lambda$: a large $\lambda$ for MAX-ENT might lead to an unstable model training process and result in a large accuracy loss. In contrast, GRL is often more stable, which is consistent to the results shown in [7].

## 5 Related Work

**Attribute Obfuscation** Various minimax formulations and algorithms have been proposed to defend against inference attack in different scenarios [4, 14, 16, 19, 22, 24, 25, 33, 34]. Bertran et al.

[4] proposed the optimization problem where the terms in the objective function are defined in terms of mutual information. Under their formulation, they analyze a trade-off between between utility loss and attribute obfuscation: under the constraint of the attribute obfuscation $I(A; Z) \leq k$, what the maximum utility loss $I(Y; X \mid Z)$ is. Compared with these works, we study the inherent trade-off between the accuracy and attribute obfuscation and provide formal guarantees to quantify worst-case inference error given the transformation.

**Differential Privacy**  Differential privacy (DP) has been proposed and extensively investigated to protect the individual privacy of collected data [9] and DP mechanisms were used in the training of deep neural network recently [1, 26]. DP ensures the output distribution of a randomized mechanism to be statistically indistinguishable between any two neighboring datasets, and provides formal guarantees for privacy problems such as defending against the membership query attacks [21, 29]. From this perspective, DP is closely related to the well-known membership inference attack [29] instead. As a comparison, our goal of attribute obfuscation is to learn a representation such that the sensitive attributes cannot be accurately inferred. Although the two goals differ, Yeom et al. [36] show the there are deep connections between membership inference and attribute inference. An interesting direction to explore is to draw more formal connections to these two notions. Last but not least, It is also worth to mention that the notion of individual fairness may be viewed as a generalization of DP [10].

**Algorithmic Fairness**  Recent work has shown that unfair models could lead to the leakage of users' sensitive information [35]. In particular, adversarial learning methods have been used as a tool in both fields to achieve the corresponding goals [12, 19]. However, the motivations and goals significantly differ between these two fields. Specifically, the widely adopted notion of group fairness, namely equalized odds [20], requires equalized false positive and false negative rates across different demographic subgroups. As a comparison, in applications where information leakage is a concern, we mainly want to ensure that adversaries cannot steal sensitive information from the data. Hence our goal is to give a worst case guarantee on the inference error that any adversary has at least to incur. To the best of our knowledge, our results in Theorem 3.1 is the first one to analyze the performance of attribute obfuscation in such scenarios. Furthermore, no prior theoretical results exist on discussing the trade-off between attribute obfuscation and accuracy under the setting of representation learning. Our proof techniques developed in this work could also be used to derive information-theoretic lower bounds in related problems as well [39, 40]. On a final note, the relationships of the above notions are visualized in Figure 3.

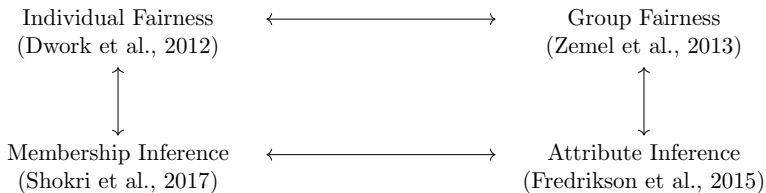

Figure 3: Relationships between different notions of fairness and inference attack.

## 6   Conclusion

We develop a theoretical framework for analyzing attribute obfuscation through adversarial representation learning. Specifically, the framework suggests using adversarial learning techniques to obfuscate the sensitive attribute and we also analyze the formal guarantees of such techniques in the limit of worst-case adversaries. We also prove an information-theoretic lower bound to quantify the inherent trade-off between accuracy and obfuscation of attribute information. Following our formulation, we conduct experiments to corroborate our theoretical results and to empirically compare different state-of-the-art attribute obfuscation algorithms. Experimental results show that the adversarial representation learning approaches are effective against attribute inference attacks and often achieve the best trade-off in terms of attribute obfuscation and accuracy maximization. We believe our work takes an important step towards better understanding the trade-off between accuracy maximization and attribute obfuscation, and it also helps inspire the future design of attribute obfuscation algorithms with adversarial learning techniques.

## Acknowledgements

HZ and GG would like to acknowledge support from the DARPA XAI project, contract #FA87501720152 and a Nvidia GPU grant. JC and YT would like to acknowledge support from NSF OAC 2002985 and NSF CNS 1943100.

## Broader Impact

In the process of data collection and information sharing, the data might contain sensitive information that the users are unwilling to disclose. This poses severe challenges for regulations such as GDPR [15] that aims to control the uses and purposes of the collected and shared data. Our work takes a step towards better understanding the trade-off therein and suggests a practical method to mitigate the potential information leakage in such high-stakes scenarios. That being said, the adversarial learning techniques might inevitably lead to degradation in target performance, and more work is needed to explore the best trade-off that could be achieved.

## Footnotes

[2]The extension of Proposition 3.1 to randomized function is staightforward as long as the randomness is independent of the sensitive attribute $A$.

[3] Some other methods [24, 31] in the literature are close variants of the above, so we do not include them here due to the space limit.

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
