[Supplementary Material]

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

## Appendix

In this appendix we provide the missing proofs of theorems and claims in our main paper. We also describe detailed experimental settings here.

## A   Technical Tools

In this section we list the lemmas and theorems used during our proof.

**Lemma A.1** (Theorem 2.2, [5]). Let $H_2^{-1}(s)$ be the inverse binary entropy function for $s \in [0,1]$, then $H_2^{-1}(s) \geq s/2 \lg(6/s)$.

**Lemma A.2** (Lin [23]). Let $\mathcal{D}$ and $\mathcal{D}'$ be two distributions, then $D_{\mathrm{JS}}(\mathcal{D}, \mathcal{D}') \leq \frac{1}{2}\|\mathcal{D} - \mathcal{D}'\|_1$.

**Theorem A.1** (Data processing inequality). Let $X \perp Y \mid Z$, then $I(X;Z) \geq I(X;Y)$.

## B   Missing Proofs

**Proposition 2.1.** Let $h_A : \mathcal{X} \to \{0,1\}$ be a hypothesis, then $\mathrm{ADV}(h_A) = 0$ iff $I(h_A(X);A) = 0$ and $\mathrm{ADV}(h_A) = 1$ iff $h_A(X) = A$ almost surely or $h_A(X) = 1 - A$ almost surely.

*Proof.* We first prove the first part of the proposition. By definition, $\mathrm{ADV}(h_A) = 0$ iff $\mathrm{Pr}_{\mathcal{D}}(h_A(X) = 1 \mid A = 1) = \mathrm{Pr}_{\mathcal{D}}(h_A(X) = 1 \mid A = 0)$, which is also equivalent to $h_A(X) \perp A$. It then follows that $h_A(X) \perp A \Leftrightarrow I(h_A(X);A) = 0$.

For the second part of the proposition, again, by definition of $\mathrm{ADV}(h_A)$, it is clear to see that we either have $\mathrm{Pr}_{\mathcal{D}}(h_A(X) = 1 \mid A = 1) = 1$ and $\mathrm{Pr}_{\mathcal{D}}(h_A(X) = 1 \mid A = 0) = 0$, or $\mathrm{Pr}_{\mathcal{D}}(h_A(X) = 1 \mid A = 1) = 0$ and $\mathrm{Pr}_{\mathcal{D}}(h_A(X) = 1 \mid A = 0) = 1$. Hence we discuss by these two cases. For ease of notation, we omit the subscript $\mathcal{D}$ from $\mathrm{Pr}_{\mathcal{D}}$ when it is obvious from the context which probability distribution we are referring to.

1. If $\mathrm{Pr}(h(X) = 1 \mid A = 1) = 1$ and $\mathrm{Pr}(h(X) = 1 \mid A = 0) = 0$, then we know that:

$$
\begin{aligned}
\mathrm{Pr}(h_A(X) \neq A) &= \mathrm{Pr}(A = 0)\mathrm{Pr}(h_A(X) \neq A \mid A = 0) + \mathrm{Pr}(A = 1)\mathrm{Pr}(h_A(X) \neq A \mid A = 1) \\
&= \mathrm{Pr}(A = 0)\mathrm{Pr}(h_A(X) = 1 \mid A = 0) + \mathrm{Pr}(A = 1)\mathrm{Pr}(h_A(X) = 0 \mid A = 1) \\
&= \mathrm{Pr}(A = 0) \cdot 0 + \mathrm{Pr}(A = 1) \cdot 0 \\
&= 0.
\end{aligned}
$$

2. If $\mathrm{Pr}(h_A(X) = 1 \mid A = 1) = 0$ and $\mathrm{Pr}(h_A(X) = 1 \mid A = 0) = 1$, similarly, we have:

$$
\begin{aligned}
\mathrm{Pr}(h_A(X) \neq 1 - A) &= \mathrm{Pr}(A = 0)\mathrm{Pr}(h_A(X) \neq 1 - A \mid A = 0) + \mathrm{Pr}(A = 1)\mathrm{Pr}(h_A(X) \neq 1 - A \mid A = 1) \\
&= \mathrm{Pr}(A = 0)\mathrm{Pr}(h_A(X) = 0 \mid A = 0) + \mathrm{Pr}(A = 1)\mathrm{Pr}(h_A(X) = 1 \mid A = 1) \\
&= \mathrm{Pr}(A = 0) \cdot 0 + \mathrm{Pr}(A = 1) \cdot 0 \\
&= 0.
\end{aligned}
$$

Combining the above two parts completes the proof. ∎

**Proposition 2.2.** If $\mathcal{H}_A$ is symmetric, then $\mathrm{ADV}(\mathcal{H}_A) + \min_{h_A \in \mathcal{H}_A} \mathrm{Pr}(h_A(X) = 0 \mid A = 1) + \mathrm{Pr}(h_A(X) = 1 \mid A = 0) = 1$.

*Proof.* By definition, we have:

$$
\begin{aligned}
1 - \mathrm{ADV}(\mathcal{H}_A) &:= 1 - \max_{h_A \in \mathcal{H}_A} \mathrm{ADV}(h_A) \\
&= \min_{h_A \in \mathcal{H}_A} 1 - \big|\mathrm{Pr}(h_A(X) = 1 \mid A = 1) - \mathrm{Pr}(h_A(X) = 1 \mid A = 0)\big| \\
&= \min_{h_A \in \mathcal{H}_A} 1 - \big(\mathrm{Pr}(h_A(X) = 1 \mid A = 1) - \mathrm{Pr}(h_A(X) = 1 \mid A = 0)\big) \\
&= \min_{h \in \mathcal{H}} \mathrm{Pr}(h_A(X) = 0 \mid A = 1) + \mathrm{Pr}(h_A(X) = 1 \mid A = 0),
\end{aligned}
$$

where the third equality holds due to the fact that $\max_{h_A \in \mathcal{H}_A} \left| \Pr(h_A(X) = 1 \mid A = 1) - \Pr(h_A(X) = 1 \mid A = 0) \right| = \max_{h_A \in \mathcal{H}_A} \left( \Pr(h_A(X) = 1 \mid A = 1) - \Pr(h_A(X) = 1 \mid A = 0) \right)$. To see this, for any specific $h_A$ such that the term inside the absolute value is negative, we can find $1 - h_A \in \mathcal{H}_A$ such that it becomes positive, due to the assumption that $\mathcal{H}_A$ is symmetric. ∎

**Proposition 3.1.** Let $f : \mathcal{X} \to \mathcal{Z}$ be a deterministic function and $\mathcal{H}_A \subseteq 2^{\mathcal{Z}}$ be a hypothesis class over $\mathcal{Z}$. For any joint distribution $\mathcal{D}$ over $X, A, Y$, if $I(f(X); A) = 0$, then $\textsc{Adv}(\mathcal{H}_A \circ f) = 0$.

*Proof.* First, by the celebrated data-processing inequality, $\forall h_A \in \mathcal{H}_A$:

$$0 \le I(h_A(f(X)); A) \le I(f(X); A) = 0.$$

By Proposition 2.1, this means that $\forall h_A \in \mathcal{H}_A$, $\textsc{Adv}(h_A) = 0$, which further implies that $\textsc{Adv}(\mathcal{H}_A \circ f) = 0$ by definition. ∎

**Lemma 3.1.** For any feature map $f : \mathcal{X} \to \mathcal{Z}$, assume that $\mathcal{H}$ contains all the probabilistic classifiers, then $\min_{h \in \mathcal{H}} \mathrm{CE}_Y(h \circ f) = H(Y \mid Z)$ and $h^*(Z) := \arg\min_{h \in \mathcal{H}} \mathrm{CE}_Y(h \circ f) = \Pr(Y = 1 \mid Z = f(X))$.

*Proof.* Let $\mathcal{D}^f$ be the induced (pushforward) distribution of $\mathcal{D}$ under the map $f : \mathcal{X} \to \mathcal{Z}$. By the definition of cross-entropy loss, we have:

$$
\begin{aligned}
\mathrm{CE}_Y(h \circ f) &= -\mathbb{E}_{\mathcal{D}} \left[ \mathbb{I}(Y = 0) \log(1 - h(f(X))) + \mathbb{I}(Y = 1) \log(h(f(X))) \right] \\
&= -\mathbb{E}_{\mathcal{D}^f} \left[ \mathbb{I}(Y = 0) \log(1 - h(Z)) + \mathbb{I}(Y = 1) \log(h(Z)) \right] \\
&= -\mathbb{E}_Z \mathbb{E}_{Y \mid Z} \left[ \mathbb{I}(Y = 0) \log(1 - h(Z)) + \mathbb{I}(Y = 1) \log(h(Z)) \right] \\
&= -\mathbb{E}_Z \left[ \Pr(Y = 0 \mid Z) \log(1 - h(Z)) + \Pr(Y = 1 \mid Z) \log(h(Z)) \right] \\
&= \mathbb{E}_Z \left[ D_{\mathrm{KL}}(\Pr(Y \mid Z) \,\|\, h(Z)) \right] + H(Y \mid Z) \\
&\ge H(Y \mid Z).
\end{aligned}
$$

It is also clear from the above proof that the minimum value of the cross-entropy loss is achieved when $h(Z)$ equals the conditional probability $\Pr(Y = 1 \mid Z)$, i.e., $h^*(Z) = \Pr(Y = 1 \mid Z = f(X))$. ∎

**Theorem 3.1.** Let $f^*$ be the optimal feature map of (3) and define $H^* := H(A \mid Z = f^*(X))$. Then for any adversary $\widehat{A}$ such that $I(\widehat{A}; A \mid Z) = 0$, $\Pr_{\mathcal{D}^{f^*}}(\widehat{A} \ne A) \ge H^* / 2 \lg(6 / H^*)$.

*Proof.* To prove this theorem, let $E$ be the binary random variable that takes value 1 iff $A \ne \widehat{A}$, i.e., $E = \mathbb{I}(A \ne \widehat{A})$. Now consider the joint entropy of $A, \widehat{A}$ and $E$. On one hand, we have:

$$H(A, \widehat{A}, E) = H(A, \widehat{A}) + H(E \mid A, \widehat{A}) = H(A, \widehat{A}) + 0 = H(A \mid \widehat{A}) + H(\widehat{A}).$$

Note that the second equation holds because $E$ is a deterministic function of $A$ and $\widehat{A}$, that is, once $A$ and $\widehat{A}$ are known, $E$ is also known, hence $H(E \mid A, \widehat{A}) = 0$. On the other hand, we can also decompose $H(A, \widehat{A}, E)$ as follows:

$$H(A, \widehat{A}, E) = H(\widehat{A}) + H(A \mid \widehat{A}, E) + H(E \mid \widehat{A}).$$

Combining the above two equalities yields

$$H(A \mid \widehat{A}, E) + H(E \mid \widehat{A}) = H(A \mid \widehat{A}).$$

Furthermore, since conditioning cannot increase entropy, we have $H(E \mid \widehat{A}) \le H(E)$, which further implies

$$H(A \mid \widehat{A}) \le H(E) + H(A \mid \widehat{A}, E).$$

Now consider $H(A \mid \widehat{A}, E)$. Since $A \in \{0, 1\}$, by definition of the conditional entropy, we have:

$$H(A \mid \widehat{A}, E) = \Pr(E = 1) H(A \mid \widehat{A}, E = 1) + \Pr(E = 0) H(A \mid \widehat{A}, E = 0) = 0 + 0 = 0.$$

To lower bound $H(A \mid \widehat{A})$, realize that

$$I(A; \widehat{A}) + H(A \mid \widehat{A}) = H(A) = I(A; Z) + H(A \mid Z).$$

Since $\widehat{A}$ is a randomized function of $Z$ such that $A \perp \widehat{A} \mid Z$, due to the celebrated data-processing inequality, we have $I(A; \widehat{A}) \leq I(A; Z)$, which implies

$$H(A \mid \widehat{A}) \geq H(A \mid Z).$$

Combine everything above, we have the following chain of inequalities hold:

$$H(A \mid Z) \leq H(A \mid \widehat{A}) \leq H(E) + H(A \mid \widehat{A}, E) = H(E),$$

which implies

$$\Pr_{\mathcal{D}^{f*}} (A \neq \widehat{A}) = \Pr_{\mathcal{D}^{f*}} (E = 1) \geq H_2^{-1}(H(A \mid Z)),$$

where $H_2^{-1}(\cdot)$ is the inverse function of the binary entropy $H(t) := -t \log t - (1-t) \log(1-t)$ when $t \in [0,1]$. To conclude the proof, we apply Lemma A.1 to further lower bound the inverse binary entropy function by

$$H_2^{-1}(H(A \mid Z)) \geq H(A \mid Z)/2 \lg(6/H(A \mid Z)),$$

completing the proof. ∎

**Lemma 3.2.** Let $\mathcal{D}_0$ and $\mathcal{D}_1$ be two distributions over $\mathcal{X}$ and let $\mathcal{D}_0^f$ and $\mathcal{D}_1^f$ be the induced distributions of $\mathcal{D}_0$ and $\mathcal{D}_1$ over $\mathcal{Z}$ by function $f$, then $d_{\mathrm{JS}}(\mathcal{D}_0^f, \mathcal{D}_1^f) \leq d_{\mathrm{JS}}(\mathcal{D}_0, \mathcal{D}_1)$.

*Proof.* Let $B$ be a uniform random variable taking value in $\{0,1\}$ and let the random variable $Z_B$ with distribution $\mathcal{D}_B^f$ (resp. $X_B$ with distribution $\mathcal{D}_B$) be the mixture of $\mathcal{D}_0^f$ and $\mathcal{D}_1^f$ (resp. $\mathcal{D}_0$ and $\mathcal{D}_1$) according to $B$. It is easy to see that $\mathcal{D}_B = (\mathcal{D}_0 + \mathcal{D}_1)/2$, and we have:

$$
\begin{aligned}
I(B; X_B) &= H(X_B) - H(X_B \mid B) \\
&= -\sum \mathcal{D}_B \log \mathcal{D}_B + \frac{1}{2} \left( \sum \mathcal{D}_0 \log \mathcal{D}_0 + \sum \mathcal{D}_1 \log \mathcal{D}_1 \right) \\
&= -\frac{1}{2} \sum \mathcal{D}_0 \log \mathcal{D}_B - \frac{1}{2} \sum \mathcal{D}_1 \log \mathcal{D}_B + \frac{1}{2} \left( \sum \mathcal{D}_0 \log \mathcal{D}_0 + \sum \mathcal{D}_1 \log \mathcal{D}_1 \right) \\
&= \frac{1}{2} \sum \mathcal{D}_0 \log \frac{\mathcal{D}_0}{\mathcal{D}_B} + \frac{1}{2} \sum \mathcal{D}_1 \log \frac{\mathcal{D}_1}{\mathcal{D}_B} \\
&= \frac{1}{2} D_{\mathrm{KL}}(\mathcal{D}_0 \| \mathcal{D}_B) + \frac{1}{2} D_{\mathrm{KL}}(\mathcal{D}_1 \| \mathcal{D}_B) \\
&= D_{\mathrm{JS}}(\mathcal{D}_0, \mathcal{D}_1).
\end{aligned}
$$

Similarly, we have:

$$D_{\mathrm{JS}}(\mathcal{D}_0^f, \mathcal{D}_1^f) = I(B; Z_B).$$

Since $\mathcal{D}_0^f$ (resp. $\mathcal{D}_1^f$) is induced by $f$ from $\mathcal{D}_0$ (resp. $\mathcal{D}_1$), by linearity, $\mathcal{D}_B^f$ is also induced by $f$ from $\mathcal{D}_B$. Hence $Z_B = f(X_B)$ and the following Markov chain holds:

$$B \to X_B \to Z_B.$$

Apply the data processing inequality, we have

$$D_{\mathrm{JS}}(\mathcal{D}_0, \mathcal{D}_1) = I(B; X_B) \geq I(B; Z_B) = D_{\mathrm{JS}}(\mathcal{D}_0^f, \mathcal{D}_1^f).$$

Taking square root on both sides of the above inequality completes the proof. ∎

**Lemma 3.3.** Let $\hat{Y} = h(f(X)) \in \{0,1\}$ be the predictor, then for $a \in \{0,1\}$, $d_{\mathrm{JS}}(\mathcal{D}_a^Y, \mathcal{D}_a^{h \circ f}) \leq \sqrt{\mathrm{Err}_a(h \circ f)}$.

*Proof.* For $a \in \{0, 1\}$, by definition of the JS distance:

$$
\begin{aligned}
d_{\text{JS}}^2(\mathcal{D}_a^Y, \mathcal{D}_a^{h \circ f}) &= D_{\text{JS}}(\mathcal{D}_a^Y, \mathcal{D}_a^{h \circ f}) \\
&\leq \|\mathcal{D}_a^Y - \mathcal{D}_a^{h \circ f}\|_1 / 2 \qquad\qquad\qquad\qquad \text{(Lemma A.2)} \\
&= (|\Pr(Y = 0 \mid A = a) - \Pr(h(f(X)) = 0 \mid A = a)| \\
&\quad\quad + |\Pr(Y = 1 \mid A = a) - \Pr(h(f(X)) = 1 \mid A = a)|) / 2 \\
&= |\Pr(Y = 1 \mid A = a) - \Pr(h(f(X)) = 1 \mid A = a)| \\
&= |\mathbb{E}[Y \mid A = a] - \mathbb{E}[h(f(X)) \mid A = a]| \\
&\leq \mathbb{E}[|Y - h(f(X))| \mid A = a] \\
&= \text{Err}_a(h \circ f),
\end{aligned}
$$

where the expectation is taken over the joint distribution of $X, Y$. Taking square root at both sides then completes the proof. ∎

**Theorem 3.2.** Let $\mathcal{H}_A \subseteq 2^{\mathcal{Z}}$ be the hypothesis space of all the classifiers from $\mathcal{Z}$ to $\{0, 1\}$. Assume the conditions in Lemma 3.4 hold, then $\forall h \in \mathcal{H}$, $\text{ACC}_0(h \circ f) + \text{ACC}_1(h \circ f) \leq 2 - \frac{1}{3} D_{\text{JS}}(\mathcal{D}_0^Y, \mathcal{D}_1^Y) + \text{ADV}(\mathcal{H}_A \circ f)$.

*Proof.* Before we delve into the details, we first give a high-level sketch of the main idea. The proof could be basically partitioned into two parts. In the first part, we will show that when $\mathcal{H}_A$ contains all the measurable prediction functions, $\text{ADV}(\mathcal{H}_A \circ f)$ could be used to upper bound $D_{\text{JS}}(\mathcal{D}_0^f, \mathcal{D}_1^f)$. The second part combines Lemma 3.3 and Lemma 3.2 to complete the proof.

In this part we first show that $D_{\text{JS}}(\mathcal{D}_0^f, \mathcal{D}_1^f) \leq \text{ADV}(\mathcal{H} \circ f)$:

$$
\begin{aligned}
D_{\text{JS}}(\mathcal{D}_0^f, \mathcal{D}_1^f) &\leq \frac{1}{2} \|\mathcal{D}_0^f - \mathcal{D}_1^f\|_1 \\
&= d_{\text{TV}}(\mathcal{D}_0^f, \mathcal{D}_1^f) \\
&= \sup_{A \in \mathscr{B}} |\mathcal{D}_0^f(A) - \mathcal{D}_1^f(A)|,
\end{aligned}
$$

where $d_{\text{TV}}(\cdot, \cdot)$ denotes the total variation distance and $\mathscr{B}$ is the sigma algebra that contains all the measurable subsets of $\mathcal{Z}$. On the other hand, when $\mathcal{H}_A$ contains all the measurable functions in $2^{\mathcal{Z}}$, we have:

$$
\begin{aligned}
\text{ADV}(\mathcal{H}_A \circ f) &= \max_{h_A \in \mathcal{H}_A} |\Pr(h_A(Z) = 1 \mid A = 0) - \Pr(h_A(Z) = 1 \mid A = 1)| \\
&= \max_{h_A \in \mathcal{H}_A} |\mathcal{D}_0(h_A^{-1}(1)) - \mathcal{D}_1(h_A^{-1}(1))| \\
&= \sup_{A \in \mathscr{B}} |\mathcal{D}_0^f(A) - \mathcal{D}_1^f(A)|,
\end{aligned}
$$

where the last equality follows from the fact that $\mathcal{H}_A$ is complete and contains all the measurable functions. Combine the above two parts we immediately have $D_{\text{JS}}(\mathcal{D}_0^f, \mathcal{D}_1^f) \leq \text{ADV}(\mathcal{H}_A \circ f)$.

Now using the key lemma, we have:

$$
\begin{aligned}
d_{\text{JS}}(\mathcal{D}_0^Y, \mathcal{D}_1^Y) &\leq d_{\text{JS}}(\mathcal{D}_0^Y, \mathcal{D}_0^{h \circ f}) + d_{\text{JS}}(\mathcal{D}_0^f, \mathcal{D}_1^f) + d_{\text{JS}}(\mathcal{D}_1^{h \circ f}, \mathcal{D}_1^Y) \\
&\leq \sqrt{\text{Err}_0(h \circ f)} + \sqrt{\text{ADV}(\mathcal{H}_A \circ f)} + \sqrt{\text{Err}_1(h \circ f)} \\
&= \sqrt{1 - \text{ACC}_0(h \circ f)} + \sqrt{\text{ADV}(\mathcal{H}_A \circ f)} + \sqrt{1 - \text{ACC}_1(h \circ f)} \\
&\leq \sqrt{3(1 - \text{ACC}_0(h \circ f) + 1 - \text{ACC}_1(h \circ f) + \text{ADV}(\mathcal{H}_A \circ f))} \\
&= \sqrt{3(2 - (\text{ACC}_0(h \circ f) + \text{ACC}_1(h \circ f) - \text{ADV}(\mathcal{H}_A \circ f)))}.
\end{aligned}
$$

Taking square at both sides and then rearrange the terms then completes the proof. ∎

**Theorem 3.3.** Assume the conditions in Theorem 3.2 hold. If $\text{ADV}(\mathcal{H}_A \circ f) \leq D_{\text{JS}}(\mathcal{D}_0^Y, \mathcal{D}_1^Y)$, then $\forall h \in \mathcal{H}$, $\text{Err}_0(h \circ f) + \text{Err}_1(h \circ f) \geq \frac{1}{2}\big(d_{\text{JS}}(\mathcal{D}_0^Y, \mathcal{D}_1^Y) - \sqrt{\text{ADV}(\mathcal{H}_A \circ f)}\big)^2$.

*Proof.* Similarly, using the key lemma, we have:

$$d_{\text{JS}}(\mathcal{D}_0^Y, \mathcal{D}_1^Y) \leq d_{\text{JS}}(\mathcal{D}_0^Y, \mathcal{D}_0^{h \circ f}) + d_{\text{JS}}(\mathcal{D}_0, \mathcal{D}_1) + d_{\text{JS}}(\mathcal{D}_1^{h \circ f}, \mathcal{D}_1^Y)$$

$$\leq \sqrt{\text{Err}_0(h \circ f)} + \sqrt{\text{ADV}(\mathcal{H}_A \circ f)} + \sqrt{\text{Err}_1(h \circ f)}$$

Under the assumption that $\text{ADV}(\mathcal{H}_A \circ f) \leq D_{\text{JS}}(\mathcal{D}_0^Y, \mathcal{D}_1^Y)$, we have $d_{\text{JS}}(\mathcal{D}_0^Y, \mathcal{D}_1^Y) \geq \sqrt{\text{ADV}(\mathcal{H}_A \circ f)}$, hence by AM-GM inequality:

$$\sqrt{2\big(\text{Err}_0(h \circ f) + \text{Err}_1(h \circ f)\big)} \geq \sqrt{\text{Err}_0(h \circ f)} + \sqrt{\text{Err}_1(h \circ f)} \geq d_{\text{JS}}(\mathcal{D}_0^Y, \mathcal{D}_1^Y) - \sqrt{\text{ADV}(\mathcal{H}_A \circ f)}.$$

Taking square at both sides then completes the proof. ∎

## C  Detailed Experiments

In this section, we provide more details of the experiments. First we provide the details of different existing methods we evaluate. Then we elaborate more dataset description, model architecture and training parameters in different experiments.

### C.1  Details on Methods

We provide a detailed description of each method here:

1). Privacy Partial Least Squares (PPLS): It learns $n \times X_d$ matrix for the feature transformation. The matrix is learned by maximizing the covariance of the learned representation and target attribute while minimizing the covariance of the learned representation and sensitive attribute.

2). Privacy Linear Discriminant Analysis (PLDA): It learns $n \times X_d$ matrix for the feature transformation. The matrix is learned by maximizing the Fisher's linear discriminability of the learned representation and target attribute while minimizing the Fisher's linear discriminability of the learned representation and sensitive attribute.

3). Minimax filter with alternative update (ALT-UP): The representation is learned via optimizing Equation 2 in an alternative way, first we update the parameters of the feature transformation module and the target attribute classifier, and then accordingly update the sensitive attribute classifier.

4). Maximum Entropy Adversarial Representation Learning (MAX-ENT): The objective equation is the slightly different from ALT-UP. The latter term contains additional entropy term to maximize unpredictability of the sensitive attribute.

5). Gradient Reversal Layer (GRL): The objective equation is the same as ALT-UP, and we train the feature transformation module by adding a gradient reversal layer between the feature transformation module and the sensitive attribute classifier.

6). Principal Component Analysis (PCA): It generates a $n \times X_d$ matrix for the feature transformation where the rows of the matrix are the $n$ largest eigenvectors of the input dataset $X$.

7). Normal Training (NORM-TRAIN): It is equivalent to normal training by setting $\lambda = 0$ in Equation 2.

8). Local Differential Privacy (LDP): Standard Laplace mechanism of local differential privacy, where the noise is added to the raw representation for erasing the information of the sensitive attribute.

9). Differentially private SGD (DPSGD): It is one of the state-of-the-art differential privacy methods on deep learning. It adds Gaussian noise to the gradients when training the model.

### C.2  Details on UCI Adult Dataset Evaluation

UCI Adult dataset is a benchmark machine learning dataset for income prediction. Each data record contains 14 categorical or numerical attributes, such as occupation, education and gender, to predict

whether individual annual income exceeds $50K/year. The dataset is divided into training set (24130 examples), validation (6032 examples), and test set (15060 examples). We choose gender, age, and education as the sensitive attributes, respectively.

Table 1: Data distribution of income ($Y$) and gender ($A$) in UCI Adult dataset.

|  | $Y = 0$ | $Y = 1$ |
|---|---|---|
| $A = 0$ | 20988 | 9539 |
| $A = 1$ | 13026 | 1669 |

Table 2: Data distribution of income ($Y$) and age ($A$) in UCI Adult dataset.

|  | $Y = 0$ | $Y = 1$ |
|---|---|---|
| $A = 0$ | 18042 | 2473 |
| $A = 1$ | 15972 | 8735 |

Table 3: Data distribution of income ($Y$) and education ($A$) in UCI Adult dataset.

|  | $Y = 0$ | $Y = 1$ |
|---|---|---|
| $A = 0$ | 20447 | 4248 |
| $A = 1$ | 13567 | 6960 |

We process each sensitive attribute as binary label for each experiment: for age label, 0 if the person is no greater than 35 years old and 1 otherwise; for education label, 0 if the person has not entered college or receive higher education than college, and 1 otherwise. In the mean time, we also remove corresponding sensitive attribute from the input, so the dimension of input data for each experiment is different. The input dimensions for income-gender experiment, income-age experiment, and income-education experiment are 113, 104 and 99, respectively. Table 1, Table 2 and Table 3 summarize the data distribution of UCI Adult dataset for protecting different sensitive attributes.

We use the two-layer ReLU-based neural net for $f$ and one-layer neural net for $h$. The output dimensions of $f$ are 64. We train all methods using SGD with the initial learning late 0.001 and momentum 0.9 for 40 epochs. In the DP-SGD experiment, we set the noise multiplier as 0.45 and 4.0 for small noise and large noise, respectively, and set the clipping norm as 1.0. $(\epsilon, \delta)$ for DPSGD small noise and DPSGD large noise are $(33.7, 10^{-5})$ and $(0.572, 10^{-5})$, respectively. Among all methods, we report the one achieving the best performance on the target task in the validation set. We run the experiments for ten random seeds and compute the average.

### C.3 Details on UTKFace Dataset Evaluation

UTKFace dataset is a large scale face dataset with annotations of age (range from 0 to 116 years old), gender (male and female), and ethnicity (White, Black, Asian, Indian, and Others). It contains 23,705 $64 \times 64$ aligned and cropped RGB face images and we split the dataset into training set (15171 examples), validation set (3793 examples) and test set (4741 examples), respectively. We further process age label and ethnicity label as binary labels: 0 if the person is not greater than 35 years old for age label (is white for ethnicity label), and 1 if the the person is greater than 35 years old for age label (is non-white for ethnicity label). Table 4 and Table 5 summarize the data distribution of UTKFace dataset for protecting different sensitive attributes.

Table 4: Data distribution of gender ($Y$) and race ($A$) in UTKFace dataset.

|  | $Y = 0$ | $Y = 1$ |
|---|---|---|
| $A = 0$ | 5477 | 4601 |
| $A = 1$ | 6914 | 6713 |

Table 5: Data distribution of gender ($Y$) and age ($A$) in UTKFace dataset.

|  | $Y = 0$ | $Y = 1$ |
|---|---|---|
| $A = 0$ | 6889 | 8218 |
| $A = 1$ | 5502 | 3096 |

Since NORM-TRAIN, ALT-UP, GRL and DP can directly enjoy the benefits of using the state-of-the-art neural network architecture as feature extraction module, so we use the feature extraction module of Wide Residual Network [37] for the (non-linear) feature transformation module, while

PPLS, PLDA, and PCA learn $12288 \times 2048$ matrix filter for $f$. We train all methods using SGD with the initial learning late 0.01 and momentum 0.9 for 30 epochs. The learning rate is decayed by a factor of 0.1 for every 10 epochs. In the DP-SGD experiment, we set the noise multiplier as 0.45 and 1.0 for small noise and large noise, respectively, and set the clipping norm as 1.0. $(\epsilon, \delta)$ for DPSGD small noise and DPSGD large noise are $(25.7, 10^{-5})$ and $(2.7, 10^{-5})$, respectively. Among all methods, we report the one achieving the best performance on the target task in the validation set. We run the experiments for ten times and compute the average.

# D  Additional Experimental Results

In this section, we present additional experimental results to gain more insights into how the trade-off parameter $\lambda$ affects the performances of different adversarial presentation learning methods. We varies the values of $\lambda$ and report the accuracies of both tasks using the Adult dataset when the sensitive attribute is gender. Note that all hyperparameter settings follow the previous experiments. The results are shown in Table 6. We can see that the overall trend is that when $\lambda$ increases, the accuracies for both tasks decrease. Compared to ALT-UP and GRL, the training of MAX-ENT is unstable when $\lambda$ is large.

Table 6: Performances of different adversarial representation learning methods when $\lambda$ changes.

| | | | 0 | 0.1 | 1 | 5 |
|---|---|---|---|---|---|---|
| Gender | ALT-UP | $\lambda$ | 0 | 0.1 | 1 | 5 |
| | | TAR. ACC. | 0.8501±0.0010 | 0.8496±0.0013 | 0.8483±0.0010 | 0.8456±0.0014 |
| | | SEN. ACC. | 0.7408±0.0096 | 0.6682±0.0026 | 0.6627±0.0021 | 0.6737±0.0005 |
| | GRL | $\lambda$ | 0 | 0.1 | 1 | 5 |
| | | TAR. ACC. | 0.8501±0.0010 | 0.8465±0.0017 | 0.8449±0.0010 | 0.8387±0.0019 |
| | | SEN. ACC. | 0.7408±0.0096 | 0.6677±0.0060 | 0.6677±0.0039 | 0.6764±0.0054 |
| | MAX-ENT | $\lambda$ | 0 | 0.1 | 1 | 5 |
| | | TAR. ACC. | 0.8501±0.0010 | 0.8450±0.0038 | 0.8411±0.0055 | 0.7891±0.0449 |
| | | SEN. ACC. | 0.7408±0.0096 | 0.6928±0.0084 | 0.6897±0.0038 | 0.5695±0.1679 |
| Age | ALT-UP | $\lambda$ | 0 | 0.1 | 1 | 5 |
| | | TAR. ACC. | 0.8467±0.0011 | 0.8468±0.0009 | 0.8472±0.0011 | 0.8451±0.0008 |
| | | SEN. ACC. | 0.7190±0.010 | 0.6516±0.0038 | 0.5422±0.0133 | 0.5573±0.0438 |
| | GRL | $\lambda$ | 0 | 0.1 | 1 | 5 |
| | | TAR. ACC. | 0.8467±0.0011 | 0.8444±0.0009 | 0.8445±0.0012 | 0.8422±0.0013 |
| | | SEN. ACC. | 0.7190±0.010 | 0.6486±0.0067 | 0.5361±0.0134 | 0.5381±0.0133 |
| | MAX-ENT | $\lambda$ | 0 | 0.1 | 1 | 5 |
| | | TAR. ACC. | 0.8467±0.0011 | 0.8379±0.0056 | 0.8194±0.0345 | 0.7795±0.0406 |
| | | SEN. ACC. | 0.7190±0.0100 | 0.6633±0.0669 | 0.6201±0.0820 | 0.5400±0.0316 |
| Education | ALT-UP | $\lambda$ | 0 | 0.1 | 1 | 5 |
| | | TAR. ACC. | 0.8494±0.0008 | 0.8498±0.0004 | 0.8497±0.0012 | 0.8494±0.0015 |
| | | SEN. ACC. | 0.7088±0.0080 | 0.6062±0.0108 | 0.6044±0.0145 | 0.5462±0.0358 |
| | GRL | $\lambda$ | 0 | 0.1 | 1 | 5 |
| | | TAR. ACC. | 0.8494±0.0008 | 0.8525±0.0010 | 0.8518±0.0007 | 0.8500±0.0013 |
| | | SEN. ACC. | 0.7088±0.0080 | 0.6082±0.0119 | 0.6015±0.0154 | 0.5528±0.0260 |
| | MAX-ENT | $\lambda$ | 0 | 0.1 | 1 | 5 |
| | | TAR. ACC. | 0.8494±0.0008 | 0.8365±0.0033 | 0.8253±0.0376 | 0.8087±0.0468 |
| | | SEN. ACC. | 0.7088±0.0080 | 0.5790±0.0383 | 0.5484±0.0001 | 0.5386±0.0305 |