[Reviews · NeurIPS 2020]

Review 1

Summary and Contributions: - The paper proposes a theoretical framework for the problem of adversarial representation learning. Specifically, the problem of obfuscating information w.r.t certain sensitive attributes, while preserving the utility (w.r.t another set of complementary attributes) of the intermediate representation. - The key contributions comprise of theoretical guarantees to bound information leakage on the sensitive attributes. - Experiments on Adult and UTKFace validate that the lower bounds hold in practise. Additionally, the authors investigate related approaches within this framework.

Strengths: 1. Unifying related methods - Many existing and related formulations towards adversarial representation learning fit within the proposed general framework. The paper additionally investigates many min-max approaches (along with DP-based approaches) in their experimental section. 2. Stepping towards rigorous guarantees - Indeed, work around adversarial representation learning is dominated by empirical obfuscation approaches, and there is little understanding on underlying theoretical aspects. The proposed theoretical results take a step towards providing rigorous guarantees. 3. Writing - The paper is overall well-written and easy to follow.

Weaknesses: 1. Novelty / Comparison with related guarantees - I appreciate that the authors intend to "bridge the gap between theory and practise". They achieve this by proving information-theoretic lower bounds on attribute predictions by adversary. - However, it appears that some works (e.g., [4] Lemma 2.1-2.3) also provide similar bounds on the information leakage. As a result, I wonder what the motivation is for the framework proposed in this paper, or how it compares to similar guarantees on quantifying attribute leakage? 2. Bounds too loose? - It is great that the bounds applies to many existing adversarial representation approaches (e.g., GRL, MAX-ENT). However, going by Fig. 1, I am concerned that the bounds might be too loose to provide meaningful guarantees. - In particular, from Fig. 1, the lower bounds (horizontal bars) are typically significantly lower than the error rates observed in practise. - Consequently, I wonder if it makes sense to rather investigate the guarantees in a more controlled setting, such as on a synthetic data. I imagine this would also alleviate the issue of being unable to "find the global optimal conditional entropy H^*" due to the non-convexity of optimizing deep networks. 3. Evaluation - I also have a concern on how the guarantees are evaluated. Specifically, I find missing the results from Fig. 1 visualized on a curve of error rate vs. hyperparameter value (or some notion of utility) to control the trade-off. I find this experiment crucial since the goal is to understand the trade-off and to additionally decouple the choices of lambda.

Correctness: They mostly appear correct.

Clarity: Yes. I liked the writing and found it easy to follow.

Relation to Prior Work: I find missing how the theoretical results on attribute leakage compares to similar works. See Concern #1 under "3. Weaknesses".

Reproducibility: Yes

Additional Feedback: =============== Post-rebuttal update =============== I will keep my score. The rebuttal addresses some of my concerns and I'm happy authors plan to make minor revisions to work on it (e.g., detailed results by varying $\lambda$, better discussion with related guarantees). I am still a bit concerned on the looseness of the bounds observed in practice (Fig. 1). Nonetheless, the work is a good step forward in presenting a common framework to study multiple representation learning methods.


Review 2

Summary and Contributions: This paper investigates the trade-off between classification accuracy (utility) and risk of privacy leakage (accuracy of feature reverse engineering attack targeting at the privacy-sensitive features). The key idea is to formulate the process of learning the policy of a data owner and that of an optimal adversary (the worst-case adversary wrt privacy protection) as a min-max game. An information-theoretic based bound concerning the trade-off learning task accuracy and the reverse engineering accuracy (attackability of the privacy-leakage attack) of the obfuscated features is given.

Strengths: It is an interesting work studying the theoretical bound of the balance between protecting data privacy by obfuscating privacy-sensitive features in a learning task and accuracy loss of the task caused by missing these features. The bound given in Theorem 3.3 is attribute-independent, it depends on the conditional distributions of the privacy-sensitive features. Furthermore, it doesn't depend on specific choices of classifiers/learners. It can provide hints on how to bound the accuracy loss theoretically given a general learner.

Weaknesses: Assuming A is binary (A =0 and A=1) limits the usage of the derived bound in practices. Though it is mentioned that the theoretical study is easy to be extended to the case where A is categorical. A Privacy-sensitive feature (features) can be numerical (scalar / vector-valued features). It is not clear how similar analysis like the bound given in Theorem 3.3 can be derived in the continuous domain.

Correctness: Yes

Clarity: Yes, it is well written

Relation to Prior Work: Yes

Reproducibility: Yes

Additional Feedback: Post-rebuttal comments: the authors' rebuttal addresses our concerns. We will keep our score and look forward to the analysis in the continuous domain.


Review 3

Summary and Contributions: In this paper, the authors proposed a new theoretical framework for attribute obfuscation to bridge the gap between the theory and the methods minimizing the potential information and preserving target accuracy. Specifically, they proposed a minimax optimization formulation to protect the given attribute and analyze its inference guarantees against worst-case adversaries. On the other hand, there is a tension between minimizing information leakage and maximizing task accuracy. To this end, they also established an information-theoretic lower bound to characterize the fundamental trade-off between accuracy and information leakage. Numerical results support their theoretical works.

Strengths: Bridge the gap between the methods and theory. Establish the information-theoretic lower bound.

Weaknesses: It only considers two classes.

Correctness: Yes

Clarity: For some equations, should put in the equation environment to highlight the important quantities.

Relation to Prior Work: Yes

Reproducibility: Yes

Additional Feedback: In line 92, the author use hat{Err}(h) to denote the empirical error of h on a sample from D, but given equation (1), I guess it should be the empirical error of h on samples S. The Lemma 3.1 reveals the connection between the original problem and the Shannon entropy. Then we are able to plug in the optimal h and h_A to obtain minimization problem (3). I was wondering whether these lemmas and theorems will still hold if we have multiple classes. Besides, I have two questions: 1. if the optimal h and h_A are always achievable, then the proposed minimax form essentially is a regularized formula; 2. if not, then a natural question is how to update your h and h_A to obtain some guaranteed results. =============== I read the rebuttal. The authors provide a straight forward approach to extend their current results to the categorical cases, in which I think the bound is sort of lose. Maybe you can add some other assumptions to derive some much tighter results. The paper is well written and organized. I will keep my score.


Review 4

Summary and Contributions: This work presents a theoretically study on the trade-off between attribute obfuscation and task accuracy. Formulating this trade-off as a minimax optimization problem, It provides a theoretical framework for attribute obfuscation and further analyses its inference guarantees against worst-case adversaries. It further proves an information-theoretic lower bound to quantify the inherent trade-off, thus provides a platform for the evaluation of existing attribute obfuscation methods.

Strengths: The proposed theoretical analysis is sound to me. The motivation is interesting and is connected, but can be distinguished to other relative studies on Privacy and fairness. The studied trade-off is a prevailing concern for real-world applications. The proposed study is valuable since it can be to evaluate the trade-off for attribute obfuscation methods and also provide insights for design new methods. The presentation of the paper is also very clear.

Weaknesses: There are other works that studies privacy or fairness from information perspective. The author may better present some analysis in the related work section.

Correctness: The proposed theoretical analysis is sound to me.

Clarity: The presentation of the paper is logical and very clear.

Relation to Prior Work: Its connection and difference between privacy and fairness are clearly discussed.

Reproducibility: Yes

Additional Feedback: I wonder your setting about representation learning with an attribute obfuscation constraint is similar to sanitizing data with respect to certain attributes in [1] and [2]. What is your connection and difference with this work in terms of both motivation and model design? I can see that you aim to formulate the problem as a trade-off between two classification problem under representation learning mechanisms. The classification error is evaluated with cross-entropy loss, i.e. Eq.(2). Is it similar to further design the task on Utility, i.e. Y. and privacy, i.e. A, to be classification with CE loss? [1] Bertran, Martin, et al. "Learning data-derived privacy preserving representations from information metrics." (2018). [2] Bertran, Martin, et al. "Adversarially learned representations for information obfuscation and inference." International Conference on Machine Learning. 2019. ============================ The rebuttal well addresses my previous concern regarding comparison with the theoretical framework in [1] and [2]. I keep my score and vote for acceptance.

[Author Response · NeurIPS 2020]

We thank all the reviewers for the time devoted to providing thoughtful suggestions. In what follows we respond to points each reviewer made individually.

**[Reviewer #1, #4 Comparison with the Theoretical Results in Other Work]**  Both our work and [4] aim to obfuscate the sensitive attribute using representation learning. However, our theoretical results are different from theirs: Bertran et al. [4] propose the optimization problem where the terms in the objective function are defined in terms of mutual information. Under their formulation, they analyze a trade-off between between utility loss and attribute obfuscation: under the constraint of the attribute obfuscation $I(A; Z) \leq k$, what the maximum utility loss ($I(Y; X \mid Z)$) is. As a comparison, our work provides a trade-off between the 0-1 loss from two groups and attribute obfuscation using attribute inference advantage and a JS-divergence term. Furthermore, we also gave a worst-case guarantee on the attribute inference error that *any* attacker has to incur, which could be used in many real-world applications to give a lower bound estimate of the attack error. We are happy to incorporate the above comparison in our next version of the paper.

[A1] is a preliminary version of [4] and shares the same setting as well as the results as [4], so it is also different from ours.

**[Reviewer #2, #3 Extension of our Results]**  As we have pointed out in our paper, our analysis could be easily extended to the categorical case: For the trade-off between accuracy and attribute obfuscation, we need to first analyze any two values that categorical variable can take on $k$ values using the similar techniques. Then we will have $C_k^2$ lower bounds, and the sum of the conditional accuracies is lower bounded by the sum of the $C_k^2$ lower bounds divided by $k - 1$. For the formal guarantee, the same proof techniques apply to the categorical case. However, our analysis does not consider the case where $A$ is continuous. We leave this as our future work.

**[Reviewer #1, Bounds Too Loose]**  To the best of our knowledge, our formal guarantee is the first one that applies to all representation learning methods under our context and our bound does not make any assumption of the underlying distribution. This means that the theoretical results are distribution-free and can be applied on any distributions. Given its wide applicability, it is possible that the bound becomes loose on one specific distribution, or a data set. That being said, it is not hard to see that the bound is tight on a degenerate distribution where $A = Y$ almost surely. On top of that, the gap between our bound and the actual error becomes even larger in experiments because we cannot guarantee to obtain the optimal $H^*$ due to the non-convex-none-concave optimization problem when training with neural nets.

**[Reviewer #1, Evaluation]**  We have performed the experiments on how the trade-off parameter $\lambda$ affects the error rate/utility and shown the results in Figure 1 and Figure 2 (see bars with different $\lambda$ values). The general trend is: with the increase of $\lambda$, both the accuracy of the target task and the ability to predict the sensitive attribute decrease. We will provide more detailed results on this experiment in the next version of our paper.

**[Reviewer #3 #4, Minor]**  With respect to the CE loss in classification: yes, actually that is what we have already used in our experiments. For Lemma 3.1, yes, this lemma will still hold in the case where we have multiple classes in our classification problem. However, it only holds for the CE loss. With respect to the achievability of the optimal $h$ and $h_A$: yes, when we can obtain the optimal ones, then the original minimax problem reduces to a minimization problem with a regularization term. To the best of our knowledge, in the most general case, i.e., non-convex-non-concave setting, there is no algorithm that can guarantee to converge to (even approximate) Nash-equilibrium. Had we have an algorithm that guarantees to converge to some approximate Nash-equilibria with gap $\gamma$, such gap $\gamma$ could be used in a straightforward way in our Theorem 3.1.

[A1] Bertran, Martin, et al. "Learning data-derived privacy preserving representations from information metrics." (2018).

[Meta-Review · NeurIPS 2020]

The paper provides novel theoretical trade-offs between attribute obfuscation and accuracy. The results are interesting and important. The reviewers have read the rebuttal and certified that their main concerns were alleviated.